# Planar Ultrametrics for Image Segmentation

**Julian Yarkony**
Experian Data Lab
San Diego, CA 92130
julian.yarkony@experian.com

**Charless C. Fowlkes**
Department of Computer Science
University of California Irvine
fowlkes@ics.uci.edu

## Abstract

We study the problem of hierarchical clustering on planar graphs. We formulate this in terms of finding the closest ultrametric to a specified set of distances and solve it using an LP relaxation that leverages minimum cost perfect matching as a subroutine to efficiently explore the space of planar partitions. We apply our algorithm to the problem of hierarchical image segmentation.

## 1 Introduction

We formulate hierarchical image segmentation from the perspective of estimating an ultrametric distance over the set of image pixels that agrees closely with an input set of noisy pairwise distances. An ultrametric space replaces the usual triangle inequality with the ultrametric inequality $d(u, v) \leq \max\{d(u, w), d(v, w)\}$ which captures the transitive property of clustering (if $u$ and $w$ are in the same cluster and $v$ and $w$ are in the same cluster, then $u$ and $v$ must also be in the same cluster). Thresholding an ultrametric immediately yields a partition into sets whose diameter is less than the given threshold. Varying this distance threshold naturally produces a hierarchical clustering in which clusters at high thresholds are composed of clusters at lower thresholds.

Inspired by the approach of [1], our method represents an ultrametric explicitly as a hierarchical collection of segmentations. Determining the appropriate segmentation at a single distance threshold is equivalent to finding a minimum-weight multicut in a graph with both positive and negative edge weights [3, 14, 2, 11, 20, 21, 4, 19, 7]. Finding an ultrametric imposes the additional constraint that these multicuts are hierarchically consistent across different thresholds. We focus on the case where the input distances are specified by a planar graph. This arises naturally in the domain of image segmentation where elements are pixels or superpixels and distances are defined between neighbors and allows us to exploit fast combinatorial algorithms for partitioning planar graphs that yield tighter LP relaxations than the local polytope relaxation often used in graphical inference [20].

The paper is organized as follows. We first introduce the closest ultrametric problem and the relation between multicuts and ultrametrics. We then describe an LP relaxation that uses a delayed column generation approach and exploits planarity to efficiently find cuts via the classic reduction to minimum-weight perfect matching [13, 8, 9, 10]. We apply our algorithm to the task of natural image segmentation and demonstrate that our algorithm converges rapidly and produces optimal or near-optimal solutions in practice.

## 2 Closest Ultrametric and Multicuts

Let $G = (V, E)$ be a weighted graph with non-negative edge weights $\theta$ indexed by edges $e = (u, v) \in E$. Our goal is to find an ultrametric distance $d_{(u,v)}$ over vertices of the graph that is close to $\theta$ in the sense that the distortion $\sum_{(u,v) \in E} \|\theta_{(u,v)} - d_{(u,v)}\|_2^2$ is minimized. We begin by reformulating this closest ultrametric problem in terms of finding a set of nested multicuts in a family of weighted graphs.

We specify a partitioning or multicut of the vertices of the graph $G$ into components using a binary vector $\bar{X} \in \{0, 1\}^{|E|}$ where $\bar{X}_e = 1$ indicates that the edge $e = (u, v)$ is "cut" and that the vertices $u$ and $v$ associated with the edge are in separate components of the partition. We use $\mathsf{MCUT}(G)$ to denote the set of binary indicator vectors $\bar{X}$ that represent valid multicuts of the graph $G$. For notational simplicity, in the remainder of the paper we frequently omit the dependence on $G$ which is given as a fixed input.

A necessary and sufficient condition for an indicator vector $\bar{X}$ to define a valid multicut in $G$ is that for every cycle of edges, if one edge on the cycle is cut then at least one other edge in the cycle must also be cut. Let $C$ denote the set of all cycles in $G$ where each cycle $c \in C$ is a set of edges and $c - \hat{e}$ is the set of edges in cycle $c$ excluding edge $\hat{e}$. We can express $\mathsf{MCUT}$ in terms of these *cycle inequalities* as:

$$\mathsf{MCUT} = \left\{ \bar{X} \in \{0, 1\}^{|E|} : \sum_{e \in c - \hat{e}} \bar{X}_e \geq \bar{X}_{\hat{e}}, \forall c \in C, \hat{e} \in c \right\} \tag{1}$$

A hierarchical clustering of a graph can be described by a nested collection of multicuts. We denote the space of valid hierarchical partitions with $L$ layers by $\bar{\Omega}_L$ which we represent by a set of $L$ edge-indicator vectors $\mathcal{X} = (\bar{X}^1, \bar{X}^2, \bar{X}^3, \ldots, \bar{X}^L)$ in which any cut edge remains cut at all finer layers of the hierarchy.

$$\bar{\Omega}_L = \{(\bar{X}^1, \bar{X}^2, \ldots \bar{X}^L) : \bar{X}^l \in \mathsf{MCUT}, \bar{X}^l \geq \bar{X}^{l+1} \; \forall l\} \tag{2}$$

Given a valid hierarchical clustering $\mathcal{X}$, an ultrametric $d$ can be specified over the vertices of the graph by choosing a sequence of real values $0 = \delta^0 < \delta^1 < \delta^2 < \ldots < \delta^L$ that indicate a distance threshold associated with each level $l$ of the hierarchical clustering. The ultrametric distance $d$ specified by the pair $(\mathcal{X}, \delta)$ assigns a distance to each pair of vertices $d_{(u,v)}$ based on the coarsest level of the clustering at which they remain in separate clusters. For pairs corresponding to an edge in the graph $(u, v) = e \in E$ we can write this explicitly in terms of the multicut indicator vectors as:

$$d_e = \max_{l \in \{0,1,\ldots,L\}} \delta^l \bar{X}_e^l = \sum_{l=0}^{L} \delta^l [\bar{X}_e^l > \bar{X}_e^{l+1}] \tag{3}$$

We assume by convention that $\bar{X}_e^0 = 1$ and $\bar{X}_e^{L+1} = 0$. Pairs $(u, v)$ that do not correspond to an edge in the original graph can still be assigned a unique distance based on the coarsest level $l$ at which they lie in different connected components of the cut specified by $X^l$.

To compute the quality of an ultrametric $d$ with respect to an input set of edge weights $\theta$, we measure the squared $L_2$ difference between the edge weights and the ultrametric distance $\|\theta - d\|_2^2$. To write this compactly in terms of multicut indicator vectors, we construct a set of weights for each edge and layer, denoted $\theta_e^l$ so that $\sum_{l=0}^{m} \theta_e^l = \|\theta_e - \delta^m\|^2$. These weights are given explicitly by the telescoping series:

$$\theta_e^0 = \|\theta_e\|^2 \qquad \theta_e^l = \|\theta_e - \delta^l\|^2 - \|\theta_e - \delta^{l-1}\|^2 \quad \forall l > 1 \tag{4}$$

We use $\theta^l \in R^{|E|}$ to denote the vector containing $\theta_e^l$ for all $e \in E$.

For a fixed number of levels $L$ and fixed set of thresholds $\delta$, the problem of finding the closest ultrametric $d$ can then be written as an integer linear program (ILP) over the edge cut indicators.

$$\min_{\mathcal{X} \in \bar{\Omega}_L} \sum_{e \in E} \left\| \theta_e - \sum_{l=0}^{L} \delta^l [\bar{X}_e^l > \bar{X}_e^{l+1}] \right\|^2 = \min_{\mathcal{X} \in \bar{\Omega}_L} \sum_{e \in E} \sum_{l=0}^{L} \|\theta_e - \delta^l\|^2 (\bar{X}_e^l - \bar{X}_e^{l+1}) \tag{5}$$

$$= \min_{\mathcal{X} \in \bar{\Omega}_L} \sum_{e \in E} \left( \|\theta_e\|^2 \bar{X}_e^0 + \sum_{l=1}^{L} \left( \|\theta_e - \delta^l\|^2 - \|\theta_e - \delta^{l-1}\|^2 \right) \bar{X}_e^l + \|\theta_e - \delta^L\|^2 \bar{X}_e^{L+1} \right)$$

$$= \min_{\mathcal{X} \in \bar{\Omega}_L} \sum_{l=0}^{L} \sum_{e \in E} \theta_e^l \bar{X}_e^l = \min_{\mathcal{X} \in \bar{\Omega}_L} \sum_{l=0}^{L} \theta^l \cdot \bar{X}^l \tag{6}$$

This optimization corresponds to solving a collection of minimum-weight multicut problems where the multicuts are constrained to be hierarchically consistent.

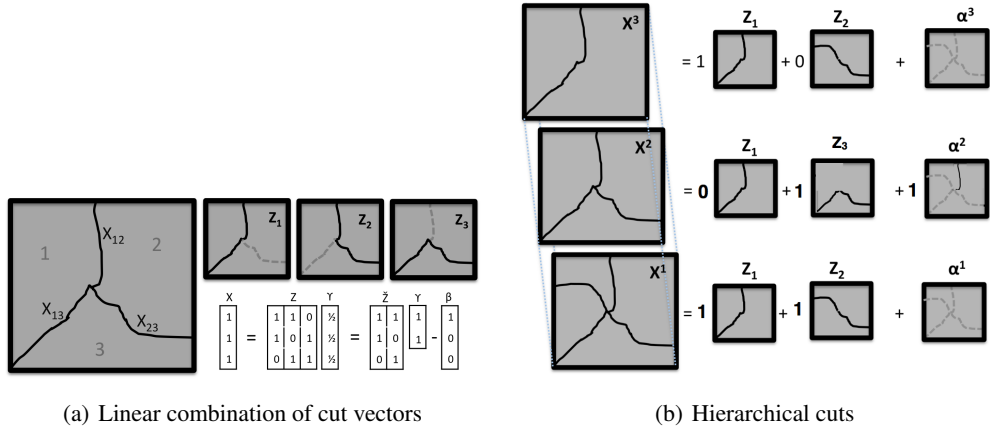

(a) Linear combination of cut vectors        (b) Hierarchical cuts

Figure 1: **(a)** Any partitioning $X$ can be represented as a linear superposition of cuts $Z$ where each cut isolates a connected component of the partition and is assigned a weight $\gamma = \frac{1}{2}$ [20]. By introducing an auxiliary slack variables $\beta$, we are able to represent a larger set of valid indicator vectors X using fewer columns of $Z$. **(b)** By introducing additional slack variables at each layer of the hierarchical segmentation, we can efficiently represent many hierarchical segmentations (here $\{X^1, X^2, X^3\}$) that are consistent from layer to layer while using only a small number of cut indicators as columns of $Z$.

Computing minimum-weight multicuts (also known as correlation clustering) is NP hard even in the case of planar graphs [6]. A direct approach to finding an approximate solution to Eq 6 is to relax the integrality constraints on $\bar{X}^l$ and instead optimize over the whole polytope defined by the set of cycle inequalities. We use $\Omega_L$ to denote the corresponding relaxation of $\bar{\Omega}_L$. While the resulting polytope is not the convex hull of MCUT, the integral vertices do correspond exactly to the set of valid multicuts [12].

In practice, we found that applying a straightforward cutting-plane approach that successively adds violated cycle inequalities to this relaxation of Eq 6 requires far too many constraints and is too slow to be useful. Instead, we develop a column generation approach tailored for planar graphs that allows for efficient and accurate approximate inference.

## 3 The Cut Cone and Planar Multicuts

Consider a partition of a planar graph into two disjoint sets of nodes. We denote the space of indicator vectors corresponding to such two-way cuts by CUT. A cut may yield more than two connected components but it can not produce every possible multicut (e.g., it can not split a triangle of three nodes into three separate components). Let $Z \in \{0, 1\}^{|E| \times |\mathsf{CUT}|}$ be an indicator matrix where each column specifies a valid two-way cut with $Z_{ek} = 1$ if and only if edge $e$ is cut in two-way cut $k$. The indicator vector of any multicut in a planar graph can be generated by a suitable linear combination of of cuts (columns of $Z$) that isolate the individual components from the rest of the graph where the weight of each such cut is $\frac{1}{2}$.

Let $\gamma \in \mathbb{R}^{|\mathsf{CUT}|}$ be a vector specifying a positive weighted combination of cuts. The set $\mathsf{CUT}^{\triangle} = \{Z\gamma : \gamma \geq 0\}$ is the conic hull of CUT or "cut cone". Since any multicut can be expressed as a superposition of cuts, the cut cone is identical to the conic hull of MCUT. This equivalence suggests an LP relaxation of the minimum-cost multicut given by

$$\min_{\gamma \geq 0} \theta \cdot Z\gamma \qquad s.t. \ Z\gamma \leq 1 \qquad (7)$$

where the vector $\theta \in \mathbb{R}^{|E|}$ specifies the edge weights. For the case of planar graphs, any solution to this LP relaxation satisfies the cycle inequalities (see supplement and [12, 18, 10]).

**Expanded Multicut Objective:** Since the matrix $Z$ contains an exponential number of cuts, Eq. 7 is still intractable. Instead we consider an approximation using a constraint set $\hat{Z}$ which is a subset

of columns of $Z$. In previous work [20], we showed that since the optimal multicut may no longer lie in the span of the reduced cut matrix $\hat{Z}$, it is useful to allow some values of $\hat{Z}\gamma$ exceed 1 (see Figure 1(a) for an example).

We introduce a slack vector $\beta \geq 0$ that tracks the presence of any "overcut" edges and prevents them from contributing to the objective when the corresponding edge weight is negative. Let $\theta_e^- = \min(\theta_e, 0)$ denote the non-positive component of $\theta_e$. The expanded multi-cut objective is given by:

$$\min_{\substack{\gamma \geq 0 \\ \beta \geq 0}} \theta \cdot \hat{Z}\gamma - \theta^- \cdot \beta \qquad s.t. \ \ \hat{Z}\gamma - \beta \leq 1 \tag{8}$$

For any edge $e$ such that $\theta_e < 0$, any decrease in the objective from overcutting by an amount $\beta_e$ is exactly compensated for in the objective by the term $-\theta_e^- \beta_e$.

When $\hat{Z}$ contains all cuts (i.e., $\hat{Z} = Z$) then Eq 7 and Eq 8 are equivalent [20]. Further, if $\gamma^\star$ is the minimizer of Eq 8 when $\hat{Z}$ only contains a subset of columns, then the edge indicator vector given by $X = \min(1, \hat{Z}\gamma^\star)$ still satisfies the cycle inequalities (see supplement for details).

# 4 Expanded LP for Finding the Closest Ultrametric

To develop an LP relaxation of the closest ultrametric problem, we replace the multicut problem at each layer $l$ with the expanded multicut objective described by Eq 8. We let $\gamma = \{\gamma^1, \gamma^2, \gamma^3 \ldots \gamma^L\}$ and $\beta = \{\beta^1, \beta^2, \beta^3 \ldots \beta^L\}$ denote the collection of weights and slacks for the levels of the hierarchy and let $\theta_e^{+l} = \max(0, \theta_e^l)$ and $\theta_e^{-l} = \min(0, \theta_e^l)$ denote the positive and negative components of $\theta^l$.

To enforce hierarchical consistency between layers, we would like to add the constraint that $Z\gamma^{l+1} \leq Z\gamma^l$. However, this constraint is too rigid when $Z$ does not include all possible cuts. It is thus computationally useful to introduce an additional slack vector associated with each level $l$ and edge $e$ which we denote as $\alpha = \{\alpha^1, \alpha^2, \alpha^3 \ldots \alpha^{L-1}\}$. The introduction of $\alpha_e^l$ allows for cuts represented by $Z\gamma^l$ to violate the hierarchical constraint. We modify the objective so that violations to the original hierarchy constraint are paid for in proportion to $\theta_e^{+l}$. The introduction of $\alpha$ allows us to find valid ultrametrics while using a smaller number of columns of $Z$ to be used than would otherwise be required (illustrated in Figure 1(b)).

We call this relaxed closest ultrametric problem including the slack variable $\alpha$ the *expanded closest ultrametric objective*, written as:

$$\min_{\substack{\gamma \geq 0 \\ \beta \geq 0 \\ \alpha \geq 0}} \sum_{l=1}^{L} \theta^l \cdot Z\gamma^l + \sum_{l=1}^{L} -\theta^{-l} \cdot \beta^l + \sum_{l=1}^{L-1} \theta^{+l} \cdot \alpha^l \tag{9}$$

$$s.t. \ \ Z\gamma^{l+1} + \alpha^{l+1} \leq Z\gamma^l + \alpha^l \quad \forall l < L$$
$$Z\gamma^l - \beta^l \leq 1 \quad \forall l$$

where by convention we define $\alpha^L = 0$ and we have dropped the constant $l = 0$ term from Eq 6.

Given a solution $(\alpha, \beta, \gamma)$ we can recover a relaxed solution to the closest ultrametric problem (Eq. 6) over $\Omega^L$ by setting $X_e^l = \min(1, \max_{m \geq l} (Z\gamma^m)_e)$. In the supplement, we demonstrate that for any $(\alpha, \beta, \gamma)$ that obeys the constraints in Eq 9, this thresholding operation yields a solution $\mathcal{X}$ that lies in $\Omega^L$ and achieves the same or lower objective value.

# 5 The Dual Objective

We optimize the dual of the objective in Eq 9 using an efficient column generation approach based on perfect matching. We introduce two sets of Lagrange multipliers $\omega = \{\omega^1, \omega^2, \omega^3 \ldots \omega^{L-1}\}$ and $\lambda = \{\lambda^1, \lambda^2, \lambda^3 \ldots \lambda^L\}$ corresponding to the between and within layer constraints respectively. For

**Algorithm 1** Dual Closest Ultrametric via Cutting Planes

$\hat{Z}^l \leftarrow \{\} \quad \forall l, \quad \text{residual} \leftarrow -\infty$
**while** residual $< 0$ **do**
    $\{\omega\}, \{\lambda\} \leftarrow$ Solve Eq 10 given $\hat{Z}$
    residual $= 0$
    **for** $l = 1 : L$ **do**
        $z^l \leftarrow \arg\min_{z \in \mathsf{CUT}}(\theta^l + \lambda^l + \omega^{l-1} - \omega^l) \cdot z$
        residual $\leftarrow$ residual $+ \frac{3}{2}(\theta^l + \lambda^l + \omega^{l-1} - \omega^l) \cdot z^l$
        $\{z(1), z(2), \ldots, z(M)\} \leftarrow \text{isocuts}(z^l)$
        $\hat{Z}^l \leftarrow \hat{Z}^l \cup \{z(1), z(2), \ldots, z(M)\}$
    **end for**
**end while**

notational convenience, let $\omega^0 = 0$. The dual objective can then be written as

$$\max_{\omega \geq 0, \lambda \geq 0} \sum_{l=1}^{L} -\lambda^l \cdot 1 \tag{10}$$
$$\theta^{-l} \leq -\lambda^l \quad \forall l$$
$$-(\omega^{l-1} - \omega^l) \leq \theta^{+l} \quad \forall l$$
$$(\theta^l + \lambda^l + \omega^{l-1} - \omega^l) \cdot Z \geq 0 \quad \forall l$$

The dual LP can be interpreted as finding a small modification of the original edge weights $\theta^l$ so that every possible two-way cut of each resulting graph at level $l$ has non-negative weight. Observe that the introduction of the two slack terms $\alpha$ and $\beta$ in the primal problem (Eq 9) results in bounds on the Lagrange multipliers $\lambda$ and $\omega$ in the dual problem in Eq 10. In practice these dual constraints turn out to be essential for efficient optimization and constitute the core contribution of this paper.

## 6 Solving the Dual via Cutting Planes

The chief complexity of the dual LP is contained in the constraints including $Z$ which encodes non-negativity of an exponential number of cuts of the graph represented by the columns of $Z$. To circumvent the difficulty of explicitly enumerating the columns of $Z$, we employ a cutting plane method that efficiently searches for additional violated constraints (columns of $Z$) which are then successively added.

Let $\hat{Z}$ denote the current working set of columns. Our dual optimization algorithm iterates over the following three steps: (1) Solve the dual LP with $\hat{Z}$, (2) find the most violated constraint of the form $(\theta^l + \lambda^l + \omega^{l-1} - \omega^l) \cdot Z \geq 0$ for layer $l$, (3) Append a column to the matrix $\hat{Z}$ for each such cut found. We terminate when no violated constraints exist or a computational budget has been exceeded.

**Finding Violated Constraints:** Identifying columns to add to $\hat{Z}$ is carried out for each layer $l$ separately. Finding the most violated constraint of the full problem corresponds to computing the minimum-weight cut of a graph with edge weights $\theta^l + \lambda^l + \omega^{l-1} - \omega^l$. If this cut has non-negative weight then all the constraints are satisfied, otherwise we add the corresponding cut indicator vector as an additional column of $Z$.

To generate a new constraint for layer $l$ based on the current Lagrange multipliers, we solve

$$z^l = \arg\min_{z \in \mathsf{CUT}} \sum_{e \in E} (\theta_e^l + \lambda_e^l + \omega_e^{l-1} - \omega_e^l) z_e \tag{11}$$

and subsequently add the new constraints from all layers to our LP, $\hat{Z} \leftarrow [\hat{Z}, \ z^1, \ z^2, \ \ldots \ z^L]$. Unlike the multicut problem, finding a (two-way) cut in a planar graph can be solved exactly by a reduction to minimum-weight perfect matching. This is a classic result that, e.g. provides an exact solution for the ground state of a 2D lattice Ising model without a ferromagnetic field [13, 8, 9, 10] in $O(N^{\frac{3}{2}} \log N)$ time [15].

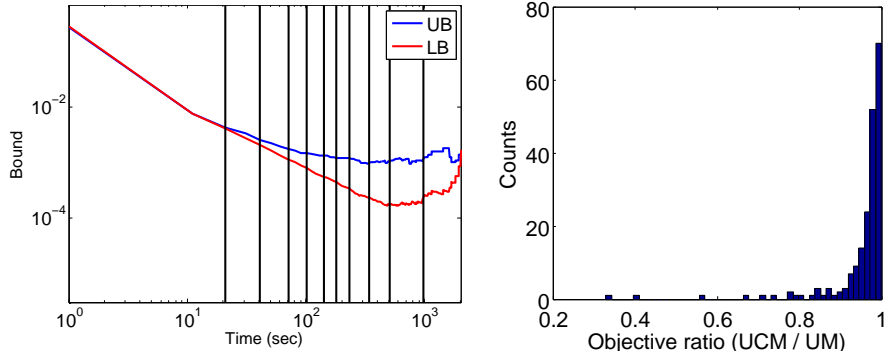

Figure 2: **(a)**: The average convergence of the upper (blue) and lower-bounds (red) as a function of running time. Values plotted are the gap between the bound and the best lower-bound computed (at termination) for a given problem instance. This relative gap is averaged over problem instances which have not yet converged at a given time point. We indicate the percentage of problem instances that have yet to terminate using black bars marking $[95, 85, 75, 65, .....5]$ percent. **(b)** Histogram of the ratio of closest ultrametric objective values for our algorithm (UM) and the baseline clustering produced by UCM. All ratios were less than $1$ showing that in no instances did UM produce a worse solution than UCM

**Computing a lower bound:** At a given iteration, prior to adding a newly generated set of constraints we can compute the total residual constraint violation over all layers of hierarchy by $\Delta = \sum_l (\theta^l + \lambda^l + \omega^{l-1} - \omega^l) \cdot z^l$. In the supplement we demonstrate that the value of the dual objective plus $\frac{3}{2}\Delta$ is a lower-bound on the relaxed closest ultrametric problem in Eq 9. Thus, as the costs of the minimum-weight matchings approach zero from below, the objective of the reduced problem over $\hat{Z}$ approaches an accurate lower-bound on optimization over $\bar{\Omega}_L$

**Expanding generated cut constraints:** When a given cut $z^l$ produces more than two connected components, we found it useful to add a constraint corresponding to each component, following the approach of [20]. Let the number of connected components of $z^l$ be denoted $M$. For each of the $M$ components then we add one column to $Z$ corresponding to the cut that isolates that connected component from the rest. This allows more flexibility in representing the final optimum multicut as superpositions of these components. In addition, we also found it useful in practice to maintain a separate set of constraints $\hat{Z}^l$ for each layer $l$. Maintaining independent constraints $\hat{Z}^1, \hat{Z}^2, \ldots, \hat{Z}^L$ can result in a smaller overall LP.

**Speeding convergence of $\omega$:** We found that adding an explicit penalty term to the objective that encourages small values of $\omega$ speeds up convergence dramatically with no loss in solution quality. In our experiments, this penalty is scaled by a parameter $\epsilon = 10^{-4}$ which is chosen to be extremely small in magnitude relative to the values of $\theta$ so that it only has an influence when no other "forces" are acting on a given term in $\omega$.

**Primal Decoding:** Algorithm 1 gives a summary of the dual solver which produces a lower-bound as well as a set of cuts described by the constraint matrices $\hat{Z}^l$. The subroutine *isocuts*$(z^l)$ computes the set of cuts that isolate each connected component of $z^l$. To generate a hierarchical clustering, we solve the primal, Eq 9, using the reduced set $\hat{Z}$ in order to recover a fractional solution $X_e^l = \min(1, \max_{m \geq l} (\hat{Z}^m \gamma^m)_e)$. We use an LP solver (IBM CPLEX) which provides this primal solution "for free" when solving the dual in Alg. 1.

We round the fractional primal solution $X$ to a discrete hierarchical clustering by thresholding: $\bar{X}_e^l \leftarrow [X_e^l > t]$. We then repair (uncut) any cut edges that lie inside a connected component. In our implementation we test a few discrete thresholds $t \in \{0, 0.2, 0.4, 0.6, 0.8\}$ and take that threshold that yields $\bar{X}$ with the lowest cost. After each pass through the loop of Alg. 1 we compute these upper-bounds and retain the optimum solution observed thus far.

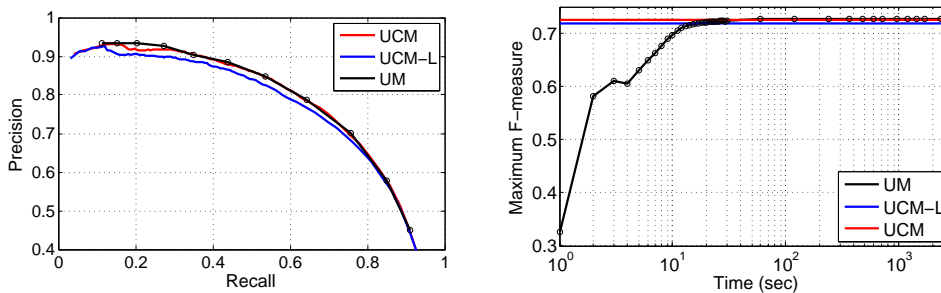

Figure 3: **(a)** Boundary detection performance of our closest ultrametric algorithm (UM) and the baseline ultrametric contour maps algorithm with (UCM) and without (UCM-L) length weighting [5] on BSDS. Black circles indicate thresholds used in the closest UM optimization. **(b)** Anytime performance: F-measure on the BSDS benchmark as a function of run-time. UM, UCM with and without length weighting achieve a maximum F-measure of 0.728, 0.726, and 0.718 respectively.

## 7 Experiments

We applied our algorithm to segmenting images from the Berkeley Segmentation Data set (BSDS) [16]. We use superpixels generated by performing an oriented watershed transform on the output of the global probability of boundary (gPb) edge detector [17] and construct a planar graph whose vertices are superpixels with edges connecting neighbors in the image plane whose base distance $\theta$ is derived from $gPb$.

Let $gPb_e$ be the local estimate of boundary contrast given by averaging the $gPb$ classifier output over the boundary between a pair of neighboring superpixels. We truncate extreme values to enforce that $gPb_e \in [\epsilon, 1 - \epsilon]$ with $\epsilon = 0.001$ and set $\theta_e = \log\left(\frac{gPb_e}{1-gPb_e}\right) + \log\left(\frac{1-\epsilon}{\epsilon}\right)$ The additive offset assures that $\theta_e \geq 0$. In our experiments we use a fixed set of eleven distance threshold levels $\{\delta_l\}$ chosen to uniformly span the useful range of threshold values $[9.6, 12.6]$. Finally, we weighted edges proportionally to the length of the corresponding boundary in the image.

We performed dual cutting plane iterations until convergence or 2000 seconds had passed. Lower-bounds for the BSDS segmentations were on the order of $-10^3$ or $-10^4$. We terminate when the total residual is greater than $-2 \times 10^{-4}$. All codes were written in MATLAB using the Blossom V implementation of minimum-weight perfect matching [15] and the IBM ILOG CPLEX LP solver with default options.

**Baseline:** We compare our results with the hierarchical clusterings produced by the Ultrametric Contour Map (UCM) [5]. UCM performs agglomerative clustering of superpixels and assigns the length-weighted averaged $gPb$ value as the distance between each pair of merged regions. While UCM was not explicitly designed to find the closest ultrametric, it provides a strong baseline for hierarchical clustering. To compute the closest $l$-level ultrametric corresponding to the UCM clustering result, we solve the minimization in Eq. 6 while restricting each multicut to be the partition at some level of the UCM hierarchy.

**Convergence and Timing:** Figure 2 shows the average behavior of convergence as a function of runtime. We found the upper-bound given by the cost of the decoded integer solution and the lower-bound estimated by the dual LP are very close. The integrality gap is typically within 0.1% of the lower-bound and never more than 1 %. Convergence of the dual is achieved quite rapidly; most instances require less than 100 iterations to converge with roughly linear growth in the size of the LP at each iteration as cutting planes are added. In Fig 2 we display a histogram, computed over test image problem instances, of the cost of UCM solutions relative to those produced by closest ultrametric (UM) estimated by our method. A ratio of less than 1 indicates that our approach generated a solution with a lower distortion ultrametric. In no problem instance did UCM outperform our UM algorithm.

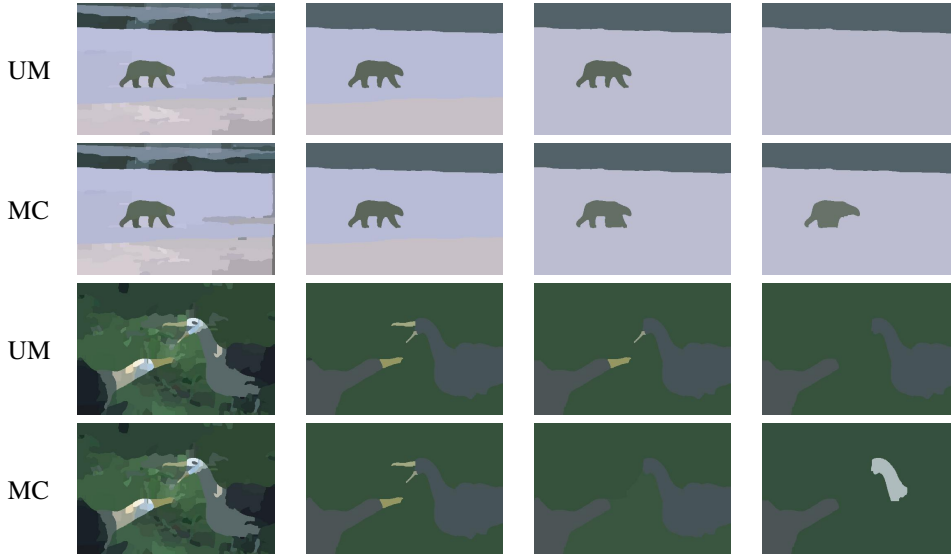

Figure 4: The proposed closest ultrametric (UM) enforces consistency across levels while performing independent multi-cut clustering (MC) at each threshold does not guarantee a hierarchical segmentation (c.f. first image, columns 3 and 4). In the second image, hierarchical segmentation (UM) better preserves semantic parts of the two birds while correctly merging the background regions.

**Segmentation Quality:** Figure 3 shows the segmentation benchmark accuracy of our closest ultrametric algorithm (denoted UM) along with the baseline ultrametric contour maps algorithm (UCM) with and without length weighting [5]. In terms of segmentation accuracy, UM performs nearly identically to the state of the art UCM algorithm with some small gains in the high-precision regime. It is worth noting that the BSDS benchmark does not provide strong penalties for small leaks between two segments when the total number of boundary pixels involved is small. Our algorithm may find strong application in domains where the local boundary signal is noisier (e.g., biological imaging) or when under-segmentation is more heavily penalized.

While our cutting-plane approach is slower than agglomerative clustering, it is not necessary to wait for convergence in order to produce high quality results. We found that while the upper and lower bounds decrease as a function of time, the clustering performance as measured by precision-recall is often nearly optimal after only ten seconds and remains stable. Figure 3 shows a plot of the F-measure achieved by UM as a function of time.

**Importance of enforcing hierarchical constraints:** Although independently finding multicuts at different thresholds often produces hierarchical clusterings, this is by no means guaranteed. We ran Algorithm 1 while setting $\omega_e^l = 0$, allowing each layer to be solved independently. Fig 4 shows examples where hierarchical constraints between layers improves segmentation quality relative to independent clustering at each threshold.

# 8    Conclusion

We have introduced a new method for approximating the closest ultrametric on planar graphs that is applicable to hierarchical image segmentation. Our contribution is a dual cutting plane approach that exploits the introduction of novel slack terms that allow for representing a much larger space of solutions with relatively few cutting planes. This yields an efficient algorithm that provides rigorous bounds on the quality the resulting solution. We empirically observe that our algorithm rapidly produces compelling image segmentations along with lower- and upper-bounds that are nearly tight on the benchmark BSDS test data set.

**Acknowledgements:** JY acknowledges the support of Experian, CF acknowledges support of NSF grants IIS-1253538 and DBI-1262547

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
