[Supplementary Material · NIPS_supplement.pdf]

# Supplement: Planar Ultrametrics for Image Segmentation

**Julian Yarkony**
Experian Data Lab
San Diego, CA 92130
julian.yarkony@experian.com

**Charless C. Fowlkes**
Department of Computer Science
University of California Irvine
fowlkes@ics.uci.edu

## 1  Expanded multicut objective and the cycle inequalities

In this appendix we show that for planar graphs, solving the expanded multicut optimization produces solutions that satisfy the cycle inequalities and have equivalent cost when truncated to lie in the unit hypercube. This establishes an equivalence between the expanded multicut optimization

$$\min_{\substack{\gamma \geq 0 \\ \beta \geq 0}} \theta \cdot \hat{Z}\gamma - \theta^- \cdot \beta \qquad s.t. \ \ \hat{Z}\gamma - \beta \leq 1 \tag{1}$$

and the cycle polytope relaxation

$$\min_{X \in \mathsf{CYC}} \theta \cdot X \tag{2}$$

for the case of planar graphs.

### 1.1  Multicut cone and Cycle cone

Recall that CUT and MCUT denote the set of binary indicator vectors that represent valid two-way cuts and multicuts respectively for a specified graph $G$. We denote the conic hulls of these sets by

$$\mathsf{CUT}^{\triangle} = \left\{ \sum_i X^i \gamma_i : \gamma_i \geq 0, X^i \in \mathsf{CUT} \right\} \tag{3}$$

$$\mathsf{MCUT}^{\triangle} = \left\{ \sum_i X^i \gamma_i : \gamma_i \geq 0, X^i \in \mathsf{MCUT} \right\} \tag{4}$$

$$\tag{5}$$

Finally, we denote the cone of positive vectors satisfying the cycle inequalities by:

$$\mathsf{CYC}^{\triangle} = \left\{ X \geq 0, \sum_{e \in c - \hat{e}} X_e \geq X_{\hat{e}}, \forall c \in C, \hat{e} \in c \right\} \tag{6}$$

We now state a two basic results concerning these cones.

**Proposition 1:** $\mathsf{MCUT}^{\triangle} = \mathsf{CUT}^{\triangle}$
Every cut indicator is a multicut indicator, hence $\mathsf{CUT}^{\triangle} \subset \mathsf{MCUT}^{\triangle}$. On the other hand, any multicut $X \in \mathsf{MCUT}$ can be written as a conic combination of cuts that isolate each connected component with weight $\frac{1}{2}$ so that $X = \frac{1}{2}\sum_i Z^i$ with $Z^i \in \mathsf{CUT}$ so $\mathsf{MCUT} \subset \mathsf{CUT}^{\triangle}$ and hence $\mathsf{MCUT}^{\triangle} \subset \mathsf{CUT}^{\triangle}$.

**Proposition 2:** If $G$ is planar, $\mathsf{CUT}^{\triangle} = \mathsf{CYC}^{\triangle}$
A stronger version of this result due to [1] states that for a graph $G$ containing no $K_5$ minor, the set of cycle inequalities over chordless circuits is sufficient to specify the facets of the cut polytope for $G$. See [2] (p. 434) for a detailed discussion.

## 1.2 The projected solution $\min(1, Z\gamma)$ satisfies the cycle inequalities

As a result of the basic properties of the cut cone, for any $\gamma \geq 0$, we have $Z\gamma \in \mathsf{CYC}^{\triangle}$ for planar graphs. Let $X = \min(1, Z\gamma)$ be a solution to the expanded multicut objective and $(Z\gamma)_e$ denote the value for a particular edge $e$. It must then be that $X \in \mathsf{CYC}^{\triangle}$ since:

$$\sum_{e \in c - \hat{e}} \min(1, (Z\gamma)_e) \geq \min(1, \sum_{e \in c - \hat{e}} (Z\gamma)_e) \tag{7}$$

$$\geq \min(1, (Z\gamma)_{\hat{e}}) \quad \forall c \in C, \hat{e} \in c \tag{8}$$

The first inequality arises from pulling the min outside the sum. The second inequality holds since $Z\gamma \in \mathsf{CYC}^{\triangle}$

## 1.3 The projected solution $\min(1, Z\gamma)$ achieves an objective cost no greater than that of $Z\gamma$

We now demonstrate that the fractional multicut $X = \min(1, Z\gamma)$ given by projecting the solution $Z\gamma$ yields a solution with an equal or smaller objective value.

Recall that $\beta$ is a positive slack variable that allows corresponding edge indicators to take on a value greater than 1.

$$Z\gamma - \beta \leq 1 \tag{9}$$

Since the objective is non-decreasing in $\beta$, for a given setting of $\gamma$ an optimal setting of the slack variables is given by:

$$\beta^* = \max(0, Z\gamma - 1) \tag{10}$$

We split the objective into positive and negative edges and write:

$$\theta \cdot Z\gamma - \theta^- \cdot \beta = \theta^+ \cdot Z\gamma + \theta^- \cdot Z\gamma - \theta^- \cdot \beta \tag{11}$$

$$= \theta^+ \cdot Z\gamma + \theta^- \cdot \min(1, Z\gamma) \tag{12}$$

$$\geq \theta^+ \cdot \min(1, Z\gamma) + \theta^- \cdot \min(1, Z\gamma) \tag{13}$$

$$= \theta \cdot \min(1, Z\gamma) \tag{14}$$

$$= \theta \cdot X \tag{15}$$

which establishes that projecting $Z\gamma$ onto the unit cube yields a fractional multicut solution that does not increase the objective.

## 2 Expanded ultrametric objective and fractional ultrametrics

Recall the set of fractional ultrametrics is defined as follows

$$\Omega_L = \left\{ \{X^1, X^2, \dots X^L\} : X^l \in \mathsf{CYC}, X^l \geq X^{l+1} \ \forall l \right\} \tag{16}$$

In analogy with the previous appendix, we show the equivalence of the expanded ultrametric rounding problem:

$$\min_{\substack{\gamma \geq 0 \\ \beta \geq 0 \\ \alpha \geq 0}} \sum_{l=1}^{L} \theta^l \cdot Z\gamma^l + \sum_{l=1}^{L} -\theta^{-l} \cdot \beta^l + \sum_{l=1}^{L-1} \theta^{+l} \cdot \alpha^l \tag{17}$$

$$s.t. \ \ Z\gamma^{l+1} + \alpha^{l+1} \leq Z\gamma^l + \alpha^l \quad \forall l < L$$

$$Z\gamma^l - \beta^l \leq 1 \quad \forall l \tag{18}$$

with the relaxed problem:

$$\min_{\mathcal{X} \in \Omega_L} \sum_{l=1}^{L} \theta^l \cdot X^l \tag{19}$$

Given an optimal solution to the expanded ultrametric rounding problem specified by $(\gamma, \alpha, \beta)$, we produce a fractional ultrametric $H$ by the projection operation:

$$H^l = \min(1, \max_{m \geq l}(Z\gamma^m)) = \max(H^{l+1}, \min(1, (Z\gamma^l))) \tag{20}$$

We show that the resulting projection $H$ yields a valid fractional ultrametric $H \in \Omega_L$ whose cost is no greater than the cost of the corresponding solution to the expanded objective.

## 2.1 Projecting expanded solutions into $\Omega_L$

By construction, $H$ satisfies the hierarchical constraint $H^l \geq H^{l+1}$. We show that $H^l \in \mathsf{CYC}$ by induction. In the previous appendix, we established that $H^L = \min(1, Z\gamma^L) \in \mathsf{CYC}$. Observe that each $H^l$ for $l < L$ is the coordinate-wise max of $H^{l+1}$ and $\min(1, Z\gamma^l)$, both of which are in $\mathsf{CYC}$ so we only need show that $\mathsf{CYC}$ is closed under coordinate-wise maximum.

Let $X^1$ and $X^2$ be two elements of $\mathsf{CYC}$ and $X^3 = \max(X^1, X^2)$. We have $\forall c \in C, \hat{e} \in c$

$$\sum_{e \in c-\hat{e}} X_e^3 = \sum_{e \in c-\hat{e}} \max(X_e^1, X_e^2) \tag{21}$$

$$\geq \max\left(\sum_{e \in c-\hat{e}} X_e^1, \sum_{e \in c-\hat{e}} X_e^2\right) \tag{22}$$

$$\geq \max(X_{\hat{e}}^1, X_{\hat{e}}^2) = X_{\hat{e}}^3 \tag{23}$$

$$\tag{24}$$

where the first inequality arises from pulling the $\max$ outside the sum and the second because $X^1$ and $X^2$ each satisfy the cycle inequality. Hence $X^3 \in \mathsf{CYC}$.

## 2.2 The cost of $H$ is no greater than that of $\{\gamma, \alpha, \beta\}$

Fixing an optimal solution to the expanded ultrametric problem specified by $\gamma$ we first note that the optimal values of $\beta$ and $\alpha$ are given by:

$$\beta^l = \max(0, Z\gamma^l - 1) \tag{25}$$

$$\alpha^l = \max_{m \geq l}(Z\gamma^m - Z\gamma^l) \tag{26}$$

The formula for $\alpha$ can be developed by starting from layer $L$ and working down, setting $\alpha$ to the smallest possible value needed to satisfy the inter-layer constraints for a given $\gamma$.

$$\alpha^L = 0$$
$$\alpha^{L-1} = \max(0, Z\gamma^L - Z\gamma^{L-1})$$
$$\alpha^{L-2} = \max(0, Z\gamma^L - Z\gamma^{L-2}, Z\gamma^{L-1} - Z\gamma^{L-2})$$
$$\dots \tag{27}$$

Since the objective is non-decreasing in $\alpha$ and $\beta$, these values are the smallest values for which the constraints are satisfied.

Plugging in the settings of the slack variables for each layer $l$ we have:

$$\theta^l \cdot Z\gamma^l - \theta^{-l} \cdot \beta^l + \theta^{+l} \cdot \alpha^l$$
$$= (\theta^{+l} + \theta^{-l}) \cdot Z\gamma^l - \theta^{-l} \cdot \max(0, Z\gamma^l - 1) + \theta^{+l} \cdot \max_{m \geq l}(Z\gamma^m - Z\gamma^l)$$
$$= \theta^{+l} \cdot (Z\gamma^l + \max_{m \geq l}(Z\gamma^m - Z\gamma^l)) + \theta^{-l} \cdot (Z\gamma^l - \max(0, Z\gamma^l - 1))$$
$$= \theta^{+l} \cdot \max_{m \geq l} Z\gamma^m + \theta^{-l} \cdot \min(1, Z\gamma^l)$$
$$\geq \theta^{+l} \cdot \min(1, \max_{m \geq l} Z\gamma^m) + \theta^{-l} \cdot \min(1, Z\gamma^l)$$
$$\geq \theta^{+l} \cdot \min(1, \max_{m \geq l} Z\gamma^m) + \theta^{-l} \cdot \min(1, \max_{m \geq l} Z\gamma^m)$$
$$= \theta^l \cdot H^l$$

where the second inequality holds because the max introduced is multiplied by a negative weight. Since projection can only remain the same or decrease the cost of each layer, the total objective must also be no greater than the expanded solution:

$$\sum_l \theta^l \cdot Z\gamma^l - \theta^{-l} \cdot \beta^l + \theta^{+l} \cdot \alpha^l \geq \sum_l \theta^l \cdot H^l$$

## 3 Derivation of Dual Problem

Here we give a derivation of the dual objective over the expanded ultrametric cut cone which we utilize to provide an efficient column generation approach based on perfect matching.

We introduce two sets of Lagrange multipliers $\{\omega^1 \ldots \omega^{L-1}\}$ and $\{\lambda^1 \ldots \lambda^L\}$ corresponding to the positivity constraints in Eq (8) in the main paper.

$$\min_{\substack{\gamma \geq 0 \\ \beta \geq 0 \\ \alpha \geq 0}} \max_{\omega \geq 0, \lambda \geq 0} \sum_{l=1}^{L} \theta^l Z \cdot \gamma^l - \sum_{l=1}^{L} \theta^{-l} \beta^l + \sum_{l=1}^{L-1} \theta^{+l} \alpha^l \tag{28}$$

$$+ \sum_{l=1}^{L-1} \omega^l (Z \cdot \gamma^{l+1} + \alpha^{l+1} - Z\gamma^l - \alpha^l)$$

$$+ \sum_{l=1}^{L} \lambda^l (Z \cdot \gamma^l - 1 - \beta^l)$$

For notational convenience, we set $\alpha^L = 0$ and $\omega^0 = 0$. We reorder the terms of the Lagrangian in terms of summations over the primal variable indices.

$$\min_{\substack{\gamma \geq 0 \\ \beta \geq 0 \\ \alpha \geq 0}} \max_{\omega \geq 0, \lambda \geq 0} \sum_{l=1}^{L} -\lambda^l 1 + \sum_{l=1}^{L} (-\theta^{-l} - \lambda^l)\beta^l \tag{29}$$

$$+ \sum_{l=1}^{L} (\theta^{+l} + \omega^{l-1} - \omega^l)\alpha^l + \sum_{l=1}^{L} (\theta^l + \lambda^l + \omega^{l-1} - \omega^l) \cdot Z\gamma^l$$

Each primal variable yields a positivity constraint in the dual.

$$\max_{\omega \geq 0, \lambda \geq 0} \sum_{l=1}^{L} -\lambda^l 1 \tag{30}$$

$$s.t. \ (-\theta^{-l} - \lambda^l) \geq 0 \qquad \qquad \forall l$$

$$(\theta^{+l} - \omega^l + \omega^{l-1}) \geq 0 \qquad \qquad \forall l$$

$$(\theta^l + \lambda^l + \omega^{l-1} - \omega^l) \cdot Z \geq 0 \qquad \qquad \forall l$$

This dual LP can be interpreted as finding modification of the original edge weights $\theta^l$ so that every possible cut of each resulting graph has non-negative weight. Observe that the introduction of the two slack terms $\alpha$ and $\beta$ in the primal problem (Eq (8) in the main paper) results in bounds on the Lagrange multipliers $\lambda$ and $\omega$ in the dual problem. The constraint $(-\theta^{-l} - \lambda^l) \geq 0$ is a result of the introduction of $\beta^l$. The constraint $\omega^{l-1} - \omega^l \leq \theta^{+l}$ is a result of the introduction of $\alpha^l$. In practice these bounds turn out to be essential for efficient optimization and are a key contribution of this paper.

It is also informative to make the substitution $\mu^l = \omega^l - \omega^{l-1}$ which yields a slightly more symmetric formulation

$$\max \sum_{l=1}^{L} -\lambda^l 1 \tag{31}$$

$$s.t. \quad 0 \leq \lambda^l \leq -\theta^{-l} \qquad \forall l$$

$$0 \leq \sum_{m=1}^{l} \mu^m \qquad \forall l \tag{32}$$

$$\mu^l \leq \theta^{+l} \qquad \forall l$$

$$(\theta^l + \lambda^l - \mu^l) \cdot Z \geq 0 \qquad \forall l$$

## 4   Producing a genuine lower bound on the optimal integer solution

Consider optimizing the Lagrangian over the set of integer solutions $\mathcal{X} \in \bar{\Omega}_L$. In this case the $\alpha, \beta$ terms disappear. For a given setting of the remaining multipliers $\omega, \lambda$ we have a lower bound on the optimal integer solution given by:

$$
\begin{aligned}
L(\omega, \lambda) &= \min_{\mathcal{X} \in \bar{\Omega}_L} \sum_{l=1}^{L} (\theta^l \bar{X}^l + \omega^l(\bar{X}^{l+1} - \bar{X}^l) + \lambda^l(\bar{X}^l - 1)) \\
&= \min_{\mathcal{X} \in \bar{\Omega}_L} \sum_{l=1}^{L} (\theta^l \bar{X}^l + \omega^{l-1} \bar{X}^l - \omega^l \bar{X}^l + \lambda^l \bar{X}^l - \lambda^l 1) \\
&= \min_{\mathcal{X} \in \bar{\Omega}_L} \sum_{l=1}^{L} (\theta^l + \omega^{l-1} - \omega^l + \lambda^l) \bar{X}^l - \lambda^l 1 \\
&= \sum_{l=1}^{L} -\lambda^l 1 + \min_{\mathcal{X} \in \bar{\Omega}_L} \sum_{l=1}^{L} (\theta^l + \omega^{l-1} - \omega^l + \lambda^l) \bar{X}^l \\
&\geq \sum_{l=1}^{L} -\lambda^l 1 + \sum_{l=1}^{L} \min_{X^l \in \mathsf{MCUT}} (\theta^l + \omega^{l-1} - \omega^l + \lambda^l) \bar{X}^l \\
&\geq \sum_{l=1}^{L} -\lambda^l 1 + \sum_{l=1}^{L} \frac{3}{2} \min_{\bar{X}^l \in \mathsf{CUT}} (\theta^l + \omega^{l-1} - \omega^l + \lambda^l) \bar{X}^l \tag{33}
\end{aligned}
$$

where the first inequality arises from dropping the constraints between layers of the hierarchy and the second inequality holds for planar graphs where the the optimal multi-cut is bounded below by $\frac{3}{2}$ the value of the optimal two-way cut (see [3]).