[Reviews · NeurIPS 2015]

Submitted by Assigned_Reviewer_1

The paper generalizes the LP-relaxation approach to correlation clustering (multicut) to produce hierarchical clustering/segmentation. This can also be understood as fitting an ultrametric into the input distance data (which is called, sligtly confusingly, ultrametric rounding). The additional constraint is that at a higher level of the hierarchy, edges can be only removed from the multicuts and not added.

The main contribution is a formulation of a suitable LP relaxation for this problem. The paper considers only planar graphs, which enables to employ column generation (detecting violated cutting planes) by planar max-cut / perfect matching (using Blossom 5). Besides that, the LP uses a number of tricks, using experience gained in past research on correlation clustering [6,16]. The LP is solved by CPLEX with Blossom5 plugged in.

The experiments are done on the Berkeley segmentation dataset, using superpixels as the graph vertices. The appendix/supplement gives a thorough and detailed comparison with one other algorithm (UCM, reference [2] in the supplement), which is a version of algglomerative clustering.

The experiments show that the method works well, which is not self-evident given several non-self-evident features in the LP formulation. On the practical side, the experiments suggest that the method is marginally better than UCM (slight refinements of contours that do not much contribute to evaluation statistics but can be visible) but notably slower - see e.g. Figure 3b in the main paper.

Clarity of the text is not great but acceptable. One objection is that it often refers to non-trivial results from previous works, which makes it hard to read for a reader not completely familiar with the topic.

I have one theoretical question. You write (paragraph starting at line 095) that the relaxation CC of MC is the convex hull (ie, the integral hull) of MC. That means, the relaxation of (1) alone is exact (ie, the optimal value of linear optimization over MC and CC are equal). But at the same time you seem to suggest (line 097) that the relaxation of (4) is proper (ie, not tight). That should mean that adding the hierarchy constraint X^l\ge X^{l+1} spoils this exactness - ie, linear optimization over \Omega_L is not the same as that over \bar{\Omega}_L. If so, this is theoretically interesting. Or did I miss something?

Minor comment: the term "ultrametric rounding" is confusing because it clashes with "rounding schemes" in combinatorial optimization (see, eg, https://en.wikipedia.org/wiki/Randomized_rounding). Note that the paper [1] uses the term in this sense, while your paper uses it for "fitting an ultrametric". Is the term used in enough other works, preferably in theoretical computer science? If not, please do not promote wrong terminology and consider changing the title.
Summary: The method proposed is non-trivial and theoretically interesting. Even if experiments did not confirm much practical improvement over a faster state-of-the-art method (UCM), I still suggest acceptance due to theoretical contribution and relatively thorough analysis.

Submitted by Assigned_Reviewer_2

PAPER SUMMARY

The paper presents a method to obtain a hierarchical clustering of a planar graph by posing the problem as that of approximating a set of edge weights using an ultrametric. This is accomplished by minimizing the \ell_2 norm between the given edge weights and the learnt ultrametric. Learning the ultrametric amounts to estimating a collection of multicuts that satisfies a hierarchical partitioning constraint. An efficient algorithm is presented that solves an approximation based on a finding a linear combination of a subset of possible two-way cuts of the graph.

COMMENTS

- How are the values of \delta^i (line 75) obtained? How crucial is this choice to the performance of the proposed method?

- How does the proposed method compare to the following related work? Is it faster? More accurate?

Kolmogorov, A faster algorithm for computing the principal sequence of partitions of a graph, Algorithmica 2010.

Please note that the paper talks explicitly about speeding-up the computation for planar graphs, which are considered in this submission.

- How does the proposed method compare to the randomized algorithm proposed in [1] (which appears to be the most closely related work)? Again, is it faster or more accurate?

- In the experiments section, no reference is provided for the baseline used (UCM). It is referred to as "ultrametric contour maps" in the caption of figure 2, and as "standard agglomerative clustering" in the caption of figure 3. Also, how does UCM+L differ from UCL. Please provide details of your baselines.

- Comparison with important baselines that are commonly used in computer vision are missing. For example,

Sharon et al., Hierarchy and adaptivity in segmenting visual scenes. Nature 2006

Arbalaez et al., Contour detection and hierarchical image segmentation.

- Minor: The paper contains several typos, and a missing reference in footnote 1, which should be corrected in subsequent versions of the paper.
Summary: It's not clear whether the proposed method is better in terms of accuracy or time complexity than the existing work on hierarchical partitioning of (planar) graphs, e.g. [1] and Kolmogorov, Algorithmica 2010. Important baselines for hierarchical segmentation are missing in the experiments.

Submitted by Assigned_Reviewer_3

The purpose of this paper is to provide an efficient algorithm for the unsupervised classification of a planar graph.

The authors first define a valid hierarchical clustering and an associated optimality criterion that accounts for the values of the edges. Then they propose a relaxed version of the problem, that they show how to solve efficiently.

Summary: A nice and easy to read paper, with a clever and efficient representation of the problem. Unfortunately far from my domain of expertise so I can absolutely not evaluate the novelty.

Author Feedback
Author rebuttal: We thank all of our reviewers for their helpful suggestions: We now address each reviewer.

Reviewer 1: We thank you for your kind words and your excitement abut our work. It is challenging because of space to include a wider discussion of the previous results however we can add a longer discussion of the relevant non-trivial results such as the Barahona work to the supplement.

We thank the reviewer for carefully observing the non-trivial alterations to the LP. The most crucial of these are the insertion of alpha and beta. If either is missing the method the relaxation requires far too many cutting planes to converge, even for a single threshold!

It is not the hierarchy constraint that makes the relaxation loose. As shown in the supplement, our relaxation for multi-cut is equivalent to optimizing the objective over the polytope defined by the cycle inequalities. However, even with a single layer (a single multi-cut or correlation clustering problem) the relaxation is loose since the cycle polytope contains non integral vertices. In short, cycle inequalities are sufficient to capture the CUT polytope in planar graphs, but not the MULTICUT polytope. We will revise the paper / supplement to make this clearer. However it should be observed that in practice for problem instances in computer vision the relaxation we employ is tight or nearly tight

We are happy to alter the title and content to remove the term rounding with regards to ultra-metrics and preserve its use when discussing LP relaxations.

Reviewer 2: We thank reviewer 2 for their careful suggestions.

We selected the values of delta empirically to span a useful range of edge strengths and sample the precision-recall curve at a reasonable density and over the whole range of recall. We found performance is not sensitive to the precise choice of delta.

We thank the reviewer for brining the paper of Kolmagorov to our attention. We note that the objective there is fundamentally different from ours and only considers minimum weight cuts of graphs with positive edges balanced by a term that counts the number of clusters, a problem that can be solved in polynomial time. Our multicut approach allows for both attractive and repulsive potentials between nodes (also known as correlation clustering) which makes it more powerful (but also NP hard). We can include a brief discussion of this in the revised paper and also plan to implement and benchmark the results.

The L in UCM+L is simply UCM plus length weighting. There is not algorithm in this domain called UCL.

Our algorithm can be compared to 1 in additional experiments that we will happily include but we are highly limited in space. Our algorithm is based on planarity and leverages the power of computing the ground state of a 2D ising model without a field. This is a key distinction of our approach.

We can do additional comparisons. However we do compare to "Arbalaez et al., Contour detection and hierarchical image segmentation.". The algorithm that the Arbalaez work operates with is UCM. UCM is the baseline grouping method. It is referenced as [2] in the appendix. We will properly cite it in the main paper. UCM is however simply a variant of agglomerative clustering.

Reviewer 3: To the best of our knowledge there is no strongly related work except for [6]

Reviewer 4: We agree that the improvement is segmentation accuracy is limited. However, the benefits over greedy merging occur in places

where there is a small "leak" in the boundary between segments that are then prematurely merged together. This is not well measured by the benchmark which counts the number of missing boundary pixels rather than segment overlap. We will include a figure which highlights this type of error in the final version of the paper.

Reviewer 5: Segmentation has been quite useful for assisting in object detection. For example the top performing RCNN object detector uses as a processing step a region proposal mechanism based on bottom-up segmentation. We can add in some discussion of such results to highlight the value of improving segmentation.

Reviewer 6: We emphasize the mathematical novelty of the approach which is significantly and non-trivially different from [6]. Like [6] we are agnostic to the features and super-pixels used. We can also employ structured learning in non-trivial ways unlike UCM. Timing can be improved by more intelligently using the LP solver.